# SGLT-2 Inhibitors in NAFLD: Expanding Their Role beyond Diabetes and Cardioprotection

**DOI:** 10.3390/ijms23063107

**Published:** 2022-03-13

**Authors:** Theodoros Androutsakos, Narjes Nasiri-Ansari, Athanasios-Dimitrios Bakasis, Ioannis Kyrou, Efstathios Efstathopoulos, Harpal S. Randeva, Eva Kassi

**Affiliations:** 1Department of Pathophysiology, Medical School, National and Kapodistrian University of Athens, 11527 Athens, Greece; t_androutsakos@yahoo.gr (T.A.); th.bacasis@gmail.com (A.-D.B.); 2Unit of Molecular Endocrinology, Department of Biological Chemistry, Medical School, National and Kapodistrian University of Athens, 11527 Athens, Greece; nnasiri@med.uoa.gr; 3Warwickshire Institute for the Study of Diabetes, Endocrinology and Metabolism (WISDEM), University Hospitals Coventry and Warwickshire NHS Trust, Coventry CV2 2DX, UK; kyrouj@gmail.com; 4Warwick Medical School, University of Warwick, Coventry CV4 7AL, UK; 5Laboratory of Dietetics and Quality of Life, Department of Food Science and Human Nutrition, School of Food and Nutritional Sciences, Agricultural University of Athens, 11855 Athens, Greece; 6Centre for Sport, Exercise and Life Sciences, Research Institute for Health & Wellbeing, Coventry University, Coventry CV1 5FB, UK; 7Aston Medical School, College of Health and Life Sciences, Aston University, Birmingham B4 7ET, UK; 82nd Department of Radiology, Medical School, National and Kapodistrian University of Athens, 11527 Athens, Greece; stathise@med.uoa.gr; 9Endocrine Oncology Unit, 1st Department of Propaedeutic Internal Medicine, Laiko Hospital, National and Kapodistrian University of Athens, 11527 Athens, Greece

**Keywords:** non-alcoholic fatty liver disease, NAFLD, MAFLD, SGLT-2, sodium-glucose co-transporter type-2 inhibitors, metabolic syndrome

## Abstract

Non-alcoholic fatty liver disease (NAFLD) is an ‘umbrella’ term, comprising a spectrum ranging from benign, liver steatosis to non-alcoholic steatohepatitis, liver fibrosis and eventually cirrhosis and hepatocellular carcinoma. NAFLD has evolved as a major health problem in recent years. Discovering ways to prevent or delay the progression of NAFLD has become a global focus. Lifestyle modifications remain the cornerstone of NAFLD treatment, even though various pharmaceutical interventions are currently under clinical trial. Among them, sodium-glucose co-transporter type-2 inhibitors (SGLT-2i) are emerging as promising agents. Processes regulated by SGLT-2i, such as endoplasmic reticulum (ER) and oxidative stress, low-grade inflammation, autophagy and apoptosis are all implicated in NAFLD pathogenesis. In this review, we summarize the current understanding of the NAFLD pathophysiology, and specifically focus on the potential impact of SGLT-2i in NAFLD development and progression, providing current evidence from in vitro, animal and human studies. Given this evidence, further mechanistic studies would advance our understanding of the exact mechanisms underlying the pathogenesis of NAFLD and the potential beneficial actions of SGLT-2i in the context of NAFLD treatment.

## 1. Introduction

Non-alcoholic fatty liver disease (NAFLD) has become a major health problem worldwide with an increasing prevalence ranging from 13% in Africa to 42% in South-East Asia [1,2]. The term NAFLD includes a variety of diseases, ranging from liver fat deposition in more than 5% of hepatocytes (steatosis—non-alcoholic fatty liver (NAFL)) to necroinflammation and fibrosis (non-alcoholic steatohepatitis (NASH)), which can progress into NASH-cirrhosis, and eventually to hepatocellular carcinoma [3,4]. Determining ways to delay the progression of NAFLD has become a global focus. 

Lifestyle modifications—namely improved diet, weight management and increased physical activity—play a fundamental role in the treatment in NAFLD, since more than half of the patients with NAFLD have a high body mass index (BMI) [5,6]. Of note, drugs for specifically treating NAFLD are now being developed/approved, thus management of co-morbidities such as obesity, dyslipidemia or type 2 diabetes mellitus (T2DM) remains the cornerstone for the treatment of NAFLD patients [7,8].

Interestingly, since the cardiometabolic disorders associated with NAFLD account for the increased morbidity and mortality in these patients, the term metabolic (dysfunction) associated fatty liver disease (MAFLD) has been recently proposed by a panel of international experts as more appropriate, reflecting the underlying pathogenesis of the NAFLD spectrum [9,10,11]. Masarone et al., revealed that the prevalence of NAFLD was 94.82% among patients with metabolic syndromes (MS) and was presented in all T2DM patients with elevated transaminases when they performed biopsy [12]. Surprisingly, 58.52% of MS and 96.82% of T2DM patients in this study were diagnosed with NASH. As insulin resistance (IR) is crucially linked with both T2DM and NASH pathophysiology [13], they concluded that NASH may be one of the early clinical manifestations of T2DM [12]. Of note, NAFLD, particularly steatohepatitis, has also been associated with an increased risk of cardiovascular-related mortality regardless of age, sex, smoking habit, cholesterolemia and the remaining elements of MS [11]. Several prospective, observational and cross-sectional studies and meta-analyses demonstrated that NAFLD is associated with enhanced preclinical atherosclerotic damage as well as coronary, cerebral and peripheral vascular events with a negative impact on patients’ outcome [14]. Furthermore, the severity of liver biopsy-based fibrosis has been independently associated with the worsening of both systolic and diastolic cardiac dysfunction [15,16].

Sodium-glucose co-transporter type-2 (SGLT-2) inhibitors (SGLT-2i) are glucose-lowering agents that improve glucose control, whilst promoting weight loss and lowering serum uric acid levels. These agents have shown great advantages even in patients with no diabetes, gaining approval of use in non-diabetic patients with heart failure and chronic kidney diseases [17,18]. In addition to the anti-hyperglycemic effects and their ability to reduce body weight, SGLT-2i seem to exert potent antioxidant and anti-inflammatory effects making them promising candidates for NAFLD treatment.

Indeed, recent data from animal studies and clinical trials have demonstrated beneficial effects of SGLT-2i on fatty liver accumulation, as judged by improvement of biological markers of NAFLD, as well as by imaging techniques, albeit mainly in T2DM patients.

Herein, we provide current insights into the effects of SGLT-2i on the progression of NAFLD, focusing on the underlying mechanisms of action. Endoplasmic reticulum (ER) stress, oxidative stress, low-grade inflammation, autophagy and apoptosis are among the SGLT-2i-regulated processes that have been shown to mediate the beneficial effects of SGLT-2i on NAFLD. Accordingly, we present the available evidence from in vitro, animal and human studies regarding the potential impact of SGLT-2i on NAFLD development and progression.

## 2. Overview of NAFLD Pathogenesis

The main theory concerning the pathophysiology of NAFLD has changed over time, reflecting the advances in our understanding of this multi-factorial disease. For years, the “two-hit” theory was the prevailing one. According to this theory, the pathophysiology of NAFLD consisted of a first “hit” representing the stage of simple steatosis alone (NAFL), which involves hepatocytic lipid accumulation and hepatic insulin resistance [19], as well as of a second “hit” from other factors (e.g., oxidative stress, ER stress, other injury), which was required for the development and progression of hepatic inflammation (NASH) and fibrosis. Replacing this initial theory, the “multiple parallel-hit” model [20] has been more recently proposed to better explain the complex pathogenesis and progression of NAFLD. According to this theory, as its name suggests, different amalgamations of numerous (epi)genetic and environmental factors, representing “hits”, dynamically interplay with each other, and can drive the development and progression of NAFLD. These factors include specific genetic polymorphisms and epigenetic modifications [21], features of metabolic syndrome [22,23,24,25] (western diet, lack of physical activity, central obesity, dysregulation of adipoknes and IR), lipotoxicity [26,27], dysbiosis of the gut microbiota [26], dysregulation of autophagy and mitochondrial function [28,29,30], ER stress [31], hepatocyte homeostasis and death [32,33], as well as inflammatory and fibrotic responses [34,35]. Notably, when the hepatic capacity to handle the primary metabolic energy substrates is overwhelmed, toxic lipid species accumulate in the liver, leading to hepatocyte dysfunction and apoptosis, along with metabolically triggered inflammation and subsequent fibrosis [36].

The hallmark of NAFLD pathogenesis seems to be an increased adipocyte-like (dys)function of the hepatocytes, when the capacity of adipose tissue to store excess energy from the diet is diminished [37,38,39,40]. In conditions of energy surplus, increased metabolic substrates in tissues [27] drive hepatic de novo lipogenesis [41,42,43], while enhancing IR leading to a vicious cycle. The accumulation of intrahepatic lipid levels are governed by the balance between lipids synthesis, uptake and lysis [44]. Loss of equilibrium between lipogenesis and lipolysis leads to intracellular accumulation of free fatty acids (FFA) and subsequent hepatocellular damage, IR, worsening of liver function, formation of hepatic steatosis and progression to NASH, cirrhosis, and hepatocellular carcinoma [44].

In most individuals with NAFLD, dysregulation of adipokines (e.g., leptin and adiponectin) and metabolic-induced inflammation impair insulin signaling in adipocytes [45,46]. This impairment, in turn, contributes to reduced FA uptake and accelerated lipolysis in subcutaneous adipose tissue, resulting in exces sive delivery to the liver [45,46,47]. When hepatocytes are overloaded with lipids, physiologically minor pathways of β-oxidation in the peroxisomes and the ER are upregulated, thus increasing the hepatocyte reactive oxygen species (ROS) production [41,48] and generating highly reactive aldehyde by-products [49]. This phenomenon leads to nuclear and mitochondrial DNA damage, phospholipid membrane disruption and cell death. Moreover, due to the mitochondrial dysfunction and the consequent impaired β-oxidation found in patients with NAFLD, FA are alternatively esterified and collected in lipid droplets in the ER [50,51], generating toxic lipid metabolites, such as diacylglycerols [24], ceramides [52,53], and lysophosphatidyl choline species [53], which in turn lead to hepatocyte dysfunction (lipotoxicity) [54,55] and ER stress. The adaptive homeostatic mechanism of ER stress, called the unfolded protein response (UPR) [56], is impaired in NAFLD patients and, thus further cell stress-sensors are activated, whilst inflammatory and apoptotic pathways are upregulated [57,58]. In addition, the gut microbiota seem to also play a critical role in NAFL and NASH pathogenesis. Gut-derived pathogens and damage-associated molecular patterns activate an intrahepatic inflammatory process via Toll-like receptor signaling and NLR family pyrin domain containing-3 (NLRP3) inflammasome activation [35,59,60]. Hepatic innate immune cells, including Kupffer cells and dendritic cells, as well as hepatic stellate cells (HSCs) are then activated and the liver parenchyma is progressively infiltrated by recruited neutrophils, monocytes, T-lymphocytes, and macrophages [34,61]. Subsequently, the secreted cytokines and growth factors intensify the inflammatory process and further contribute to the fibrotic process as an ineffective attempt for tissue regeneration [35]. Overall, NAFLD is characterized by a complex pathogenetic mechanism, where various pathways are implicated, making the discovery of a “wonder drug” a difficult task (Figure 1).

## 3. SGLT-2i Overview

Phlorizin, a b-Dglucoside, was the first natural non-selective SGLT-2i isolated from the bark of the apple tree in 1835 and was initially used for fever relief and treatment of infectious diseases [62,63]. Phlorizin contains a glucose moiety and an aglycone in which two aromatic carbocycles are joined by an alkyl spacer. Approximately fifty years later, scientists observed that phlorizin in high dose lowered plasma glucose levels through glucosuria independently of insulin secretion [62,63,64]. However, the mechanism of phlorizin action through the active glucose transport system of the proximal tubule was revealed in the early 1970s [62,65].

Thereafter, researchers began to discover the exact mechanism of phlorizin action and its potential application in the management of hyperglycaemia. Phlorizin, when delivered orally, undergoes hydrolysis of the O-glycosidic bond by intestinal glucosidases, leading to formation of phloretin, which is capable of GLUT transporter inhibition and uncoupling of oxidative phosphorylation. Due to Phlorizin’s low SGLT-2 selectivity, rapid degradation, inhibition of the ubiquitous glucose transporter 1 (GLUT1), poor intestinal absorption and its gastrointestinal side effects (e.g., diarrhea and dehydration), possibly due to its higher potency for SGLT-1, it failed to be developed as an antidiabetic drug [62,66,67,68].

Nevertheless, in order to overcome the aforementioned side effects of phlorizin, a researcher developed a devphlorizin-based analog with improved bioavailability/stability and more selectivity towards SGLT-2 than SGLT-1. Initially, they focused on the development of O-glucoside analogs such as T-1095, sergliflozin and remogliflozin [69,70]. However, due to their poor stability and incomplete SGLT-2 selectivity, pharmaceutical industries turned to another derivative of phlorizin known as C-glucosides [62].

Since then, there have been numerous attempts to synthesize phlorizin C-glucoside analogs with sufficient potency and selectivity for inhibition of SGLT-2. These attempts led to development of dapagliflozin by Meng et al., in 2008 [71]. Dapagliflozin demonstrated more than 1200-fold higher potency for human SGLT-2 versus SGLT-1. Apart from dapagliflozin, during the next years, several other C-glucoside inhibitors have been developed and approved by United States Food and Drug Administration (FDA). Canagliflozin, characterized by a thiophene derivative of C-glucoside, showed over 400-fold difference in inhibitory activities between human SGLT-2 and SGLT-1 [72]. Thereafter, empagliflozin, which has the highest selectivity for SGLT-2 over SGLT-1 (approximately 2700-fold), was the third agent approved by both the European Medicines Agency (EMA) and the FDA, while other SGLT-2i, namely luseogliflozin and topogliflozin, are approved so far for use only in Japan, and ipragliflozin only in Japan and Russia [73,74,75,76,77].

The structure of phlorizin and currently FDA-approved SGLT-2 inhibitors in the United States, including dapagliflozin, canagliflozin, empagliflozin, and ertugliflozin, along with their chemical formulae and brand names are presented in Figure 2 [78]. Diverse structures of other SGLT-2 inhibitors have been previously disclosed in various articles and in a number of patents [79,80].

## 4. SGLT-2 Inhibitors in NAFLD

### 4.1. Laboratory Experiments

#### 4.1.1. In Vitro Data

The expression of both SGLT-1 and SGLT-2 co-transporters has been reported in human hepatocellular carcinoma cells (HepG2) [81,82,83], whilst SGLT-2 has also been identified in immortalized human primary hepatocyte cells (HuS-E/2), as well as immortalized normal human hepatocyte-derived liver cells (L02) [84,85].

Studies have indicated that most SGLT-2i exert anti-proliferative activity in several hepatocellular cell lines, through—among other mechanisms—attenuation of glucose uptake [86]. Incubation of both L02 and HepG2 cells with canagliflozin at various concentrations for 24 h resulted in a significant reduction of cell proliferation through an increase in G0/G1 and a decrease in G2/M phase cell population [86]. In addition, an apoptotic effect was exhibited through activation of caspase 3 in HepG2 cells [87]. From a mechanistic point of view, cell growth regulators, cyclins and cyclin-dependent kinases (CDKs) have been indicated as direct targets of SGLT-2i to control proliferation and survival processes. As such, canagliflozin-treated HepG2 cells demonstrated increased expression of the cell growth regulator hepatocyte nuclear factor 4α (HNF4α) [82]. Furthermore, reduced expression of cyclin D1, cyclin D2 and cdk4 leading to cell cycle arrest in a hepatocellular carcinoma (HCC) cell line have been reported after treatment with canagliflozin [82,87]. In another study, incubation of HepG2 cells with canagliflozin resulted in reduced clonogenic cell survival and elevation of anti-carcinogenic potential of γ-irradiation through modulation of ER stress-mediated autophagy and cell apoptosis [88]. On the contrary, trilobatin, a novel SGLT-1/2 inhibitor, increased the HepG2 cell proliferation rate at a dose of 10, 50 and 100 μΜ, while incubation of human HCC cells with tofogliflozin at various concentrations did not alter the HCC cell proliferation rate [89]. Incubation of HepG2 cells with dapagliflozin, canagliflozin and empagliflozin had no effect on cancer cell survival [81,86] and adhesion capacity [81], while the cell sensitivity to dapagliflozin was induced after UDP Glucuronosyltransferase Family 1 Member A9 (UGT1A9) silencing, indicated by an increased number of floating HepG2 cells; of note, UGT1A9 metabolizes and deactivates dapagliflozin [81].

Dapagliflozin treatment remarkably suppressed oleic acid (OA)-induced lipid accumulation and TG content in L02 cells through increased FA β-oxidation, as indicated by elevated proliferator-activated receptor-gamma coactivator-1 alpha (PGC-1α) levels and activation of the AMP-activated protein kinase (AMPK)/mammalian target of rapamycin (mTOR) signaling pathway [84]. Several in vitro studies have shown that AMPK is a key regulator that mediates the various beneficial effects of SGLT-2i related to cholesterol and glucose metabolism in hepatic cells. AMPK activation by dapagliflozin prevented glucose absorption through reduced SGLT-2 expression in OA-stimulated HuS-E/2 cells [90]. This effect was eliminated after incubation of cells with compound C, a potent AMPK inhibitor [84]. Moreover, treatment of HepG2 cells with canagliflozin facilitated hepatic cholesterol efflux via activation of AMPK. In turn, AMPK activation led to increased expression of the liver X receptor (LXR) and its downstream proteins, resulting in subsequent stimulation of cholesterol reverse transport [91]. LXR activation also accelerates fecal cholesterol disposal through regulation of ATP-binding cassette (ABC) transporters ABCG5 and ABCG8 expression [92]. Canagliflozin treatment resulted in reduced expressions of ABCG5 and ABCG8 and LXR, while this effect was inhibited after treatment of cells with compound C [91]. An increased AMP/ADP ratio leads to AMPK activation. It is known that SGLT-2i reduces cellular ATP levels and indirectly activates the AMPK signaling pathway [83,93]. It has been shown that dapagliflozin alleviates the intracellular ATP levels via regulation of glucose metabolism [84].

Stimulation of hepatic cell lines with palmitic acid (PA) mimics hepatocyte situation in NASH [94,95]. Incubation of PA-stimulated L02 and HepG2 cell lines with dapagliflozin resulted in a significant reduction of intracellular lipid accumulation. This effect was attributed to down-regulation of proteins related to lipid synthesis, up-regulation of genes involved in fatty acid oxidation (e.g., peroxisome proliferator-activated receptor alpha (PPARα) and carnitine palmitoyltransferase 1 (CPT1a)), regulation of AMPK/mTOR pathway and autophagy. Interestingly, incubation of cells with compound C abolished the dapagliflozin beneficial effect on reduction of intracellular lipid accumulation [85]. Collectively, these data suggest that SGLT-2i induced improvement of NAFLD is directly dependent on AMPK signaling activation.

The divergent results of the various relevant studies suggest that the effects of SGLT-2i on cell proliferation, survival and apoptosis—which are important processes in hepatocellular carcinoma development and progression—appear to be drug-, dose- and duration-dependent.

#### 4.1.2. Animal Studies

Overall, the animal models of NAFLD/NASH can be divided into two categories. Of these, one comprises genetically modified animal models, such as leptin deficient (ob/ob), leptin-resistant (db/db), Agouti mutation (KK-Ay), and apolipoprotein E knockout (ApoE^−/−^) mice, as well as the Prague hereditary hypertriglyceridemic (HHTg) rats, Zucker diabetic fatty (ZDF) rats and Otsuka Long-Evans Tokushima Fatty (OLETF) rat. The other category includes dietary/pharmacological manipulation-induced NAFLD/NASH animal models, such as those fed with either a methionine/choline-deficient (MCD) diet, the trans-fat containing AMLN (amylin liver non-alcoholic steatohepatitis, NASH) diet, and the methionine-reduced, choline-deficient and amino acid defined (CDAA) diet, as well as a high-fat diet (HFD), and a HFD + cholesterol diet or a high-fat, high-calorie (HFHC) diet [30,96,97,98,99,100]. Most of the recent data indicate that both SGLT-1 and SGLT-2 genes are expressed in hepatic tissues of mice and rats [30,85,87,101]. High levels of SGLT-2 protein have also been detected in hepatic macrophages and T cells [101]. While the exact underlying molecular mechanisms of SGLT-2i-induced beneficial effects on NAFLD have not yet been fully clarified, most of our knowledge regarding the potential hepato-protective mechanism(s) of SGLT-2i actions comes from animal studies.

Although the SGLT-2i-induced weight loss seems to play an important role in hepato-protective effects on NAFLD in humans [102,103,104], data from animal studies suggest that other mechanisms are more likely responsible for the noted SGLT-2i-mediated hepato-protection.

Administration of SGLT-2i leads to net loss of calories and consequent attenuation of body weight gain, while reducing the accumulated white adipose tissue [99,105,106,107]. In fact, it has been found that SGLT-2i promote weight loss through improvement of systemic IR, increased body temperature and basal metabolism via regulation of the AMP/ATP ratio [19]. Petito da Silva et al., reported that empagliflozin (Table 1) at a dose of 10 mg/kg/day given for five weeks significantly reduces body weight and body mass in HFD-fed male C57Bl/6 mice, despite a slight increase in their appetite [107]. Similar results were observed in HFD-fed C57BL/6J male mice after administration of canagliflozin (Table 2) at 30 mg/kg dose for four weeks in parallel with improvement of liver function, liver TG content and NAS score [108]. Additionally, short term and low dose administration of dapagliflozin (Table 3) reduced both ALT levels and body weight in both HFD and HFD+MCD diet fed C57BL/6J mice [109]. However, it should be noted that not all studies point towards a beneficial effect of SGLT-2i on body weight. Several studies have shown that administration of SGLT-2i does not exert any significant effect on body weight gain, epididymal fat weight or food intake [30,105,110,111]. In addition, although several studies reported strong effects of low dose ipragliflozin (Table 4) on weight loss [100,112,113], supplementation of an AMLN diet with 40 mg ipragliflozin/kg for 8 weeks did not significantly alter the body weight of AMLN diet fed mice, while it improved liver function, hepatic fibrosis and NA score [114]. Empagliflozin at the dose of 10 mg/kg/day improved liver function and NAFLD status, while it had no significant effect on the body weight of ZDF, HHTg and Wistar rats [115,116], as well as on the body weight and appetite of ApoE knockout mice [30].

Although there are studies showing that administration of SGLT-2i does not exert any significant effect on body weight gain [30,100,110,111,112,113,114], the vast majority of the literature points towards a beneficial effect on weight loss.

However, the protective role of SGLT-2i against NAFLD progression also seems to be mediated by effects beyond those on body weight. Indeed, it has been shown that SGLT-2i treatment improves IR and ameliorates the intracellular FFA, total cholesterol (TC) and TG accumulation through reducing the expression of genes involved in de novo lipogenesis, FA uptake and hepatic TG secretion, whilst it also promotes the expression of key regulatory genes of fatty acid β-oxidation [30,84,100,107,109,112,117]. Specifically, empagliflozin administration has been shown to reduce hepatic triacylglycerol (TAG) levels and improve IR through reduction of lipotoxic intermediates formation, such as diacylglycerols (DAG) [115], which at elevated levels contribute to IR by activating the protein kinase C (PKC)ε pathway [118]. Of note, a recent study by Hüttl et al., demonstrated that reduction of hepatic lipid accumulation following empagliflozin treatment is mediated via increased nuclear factor erythroid 2-related factor 2 (Nrf2), fibroblast growth factor 21 (FGF21) and altered gene expression of the P450 (CYP) enzyme superfamily of cytochrome [115]. Interestingly, Nrf2 and FGF21 regulate lipid metabolism through inhibiting lipogenesis and improving insulin sensitivity, respectively [119,120].

Sterol regulatory element-binding transcription factor 1 (SREBP1) is one of the key regulators of hepatic lipogenesis and plays a crucial role in the regulation of lipogenic genes, such as fatty acid synthase (Fasn), acetyl-CoAcarboxylase 1 (Acc1), and stearoyl-CoA desaturase1 (Scd-1). Increased SREBP1 expression has been linked to the aggravation of hepatic steatosis [121]. While reduced expression of SREBP1 and Scd-1 by both canagliflozin and empagliflozin has been observed in several studies [30,107,108,115], Kern et al., found that empagliflozin treatment for eight weeks does not markedly affect the Scd-1 levels in db/db mice [122]. Dapagliflozin (1 mg/kg/day) treatment for nine weeks resulted in down-regulation of Scd-1 gene expression, as well as reduced ACC1 phosphorylation in ZDF rats. Taking into account that SREBP1 activity is directly regulated by mTOR signaling [123], reduced expression of SREBP1 and its downstream lipogenic targets could be due to the reduced mTOR expression/activity observed after SGLT-2i administration [84,123].

Peroxisome proliferator-activated receptor gamma (PPAR-γ) upregulation triggers de novo lipogenesis and consequent deposition of lipid droplets into hepatocytes [124]. Eight weeks of dapagliflozin treatment reduced hepatic weight and prevented progression of hepatic steatosis compared to mice treated with insulin glargine [125]. This effect was found to be mediated through decreased expression of PPAR-γ targeted genes involved in fatty acid synthesis, such as Scd-1, monoacylglycerol O-acyltransferase 1 (Mogat1), Cell death inducing DFFA like effector A (Cidea) and cell death inducing DFFA like effector C (Cidec), without affecting insulin sensitivity, liver TC or oleate content [125].

SGLT-2i have been also shown to reverse the HFD-induced down regulation of genes involved in FA β-oxidation and lipolysis in the liver of various mouse models of NAFLD [30,84,107,108]. Indeed, it has been shown that expression of PPARα is induced by SGLT-2i [84,105,107,114]. PPARα activation is predominant for regulation of genes related to the FA β-oxidation process in mitochondria, such as peroxisomal acyl-CoA oxidase1 (ACOX1) and enoyl-CoA hydratase, CPT1, and cytochrome-mediated (CYP4A1 and CYP4A3) [44]. Ipragliflozin treatment has been shown to result in acceleration of β-oxidation and export of very-low-density lipoprotein (VLDL) through upregulation of PPARα, CPT1A, and Microsomal Triglyceride Transfer Protein (MTTP) gene expression in the liver of AMLN-fed C57BL/6J mice. MTTP plays a crucial role in export of hepatic LDL [114]. Similarly, dapagliflozin attenuated hepatic lipid accumulation in ZDF rats via up-regulation of the FA β-oxidation enzyme ACOX1 [85]. Tofogliflozin administration induced the expression of genes related to FA β-oxidation in non-tumorous liver lesions, while it had no effect on their expression in tumorous liver tissues [126]. However, not all studies show a promotive effect of SGLT-2i on the expression of enzymes and transcription factors involved in the regulation of FA β-oxidation [107,114,126]. Of note, the expression of PPARα remained unaffected in both Wistar and HHTg rats after eight weeks of empagliflozin administration [107,115]. Administration of canagliflozin at doses of 10 and 20 mg/kg/day ameliorated the HFD-induced down-regulation of the expression of the key regulatory molecule of hepatic lipolysis, known as zinc alpha 2 glycoprotein (ZAG). When the same SGLT-2i was given in KK-Ay mice, it reduced the accumulation of lipid droplets in the liver by increasing hepatic prostaglandin E2 (PGE2) protein levels [111]. Furthermore, the combined therapy with canagliflozin and exercise exerted an additive effect on hepatic PPARγ Coactivator 1 Beta (PGC1b) elevation, and reduced expression of hepatic lipogenic genes, such as Scd1 [108]. Notably, PGC1 plays a central role in the control of hepatic gluconeogenesis, while its overexpression has been associated with stimulation of hepatic FA oxidation [127].

The inhibition of de novo lipogenesis, and enhanced lipolysis and β-oxidation by SGLT-2-i lead to a consequent reduction in oxidative stress, hepatic inflammation and apoptosis [128]. Administration of canagliflozin to HFD-fed diabetic Wistar adult male rats resulted in reduced hepatic steatosis through amelioration of oxidative stress, inflammation and apoptosis, as indicated by reduced plasma malondialdehyde (MDA) levels, serum tumor necrosis factor (TNF) and caspase-3 levels, as well as hepatic expression of interleukin-6 (IL-6) [128,129]. Moreover, dapagliflozin treatment reduced the expression of hepatic inflammatory cytokine (TNF-α, IL-1β, IL-18) content and improved hepatic steatosis in HCHF fed male Wistar rats [130]. Tahara et al., have also shown anti-oxidative stress and anti-inflammatory effects of ipragliflozin on HFD-fed and streptozotocin–nicotinamide-induced type 2 diabetic mice [131]. Similarly, dapagliflozin treatment at a dose of 1 mg/kg/day decreased hepatic ROS production of myeloperoxidase (MPO) and F4/80, thus ameliorating serum ALT levels and hepatic fibrosis [132]. MPO is a chlorinating oxidant-generating enzyme that regulates the initiation of an acute inflammatory response and promotes the development of chronic inflammation through oxidant production [133]. Furthermore, canagliflozin treatment of F344 rats for 16 weeks resulted in reduced hepatic ROS production and MDA and 8-hydroxy-2′-deoxyguanosine (8-OHdG) levels [110]. MDA and 8-OHdG are known biomarkers of lipid peroxidation and oxidative DNA damage, respectively. Of note, canagliflozin and teneligliptin combination therapy showed a stronger effect on reduction of hepatic oxidative stress and amelioration of hepatic inflammation [110]. Treatment with both remogliflozin and ipragliflozin also reduced oxidative stress levels, as evaluated by decreased thiobarbituric acid reactive substances (TBARS) levels [100,134]. Instead, administration of dapagliflozin at 1 mg/kg/twice/day for four weeks had no significant effect either on hepatic TBARS and TG levels or on plasma ALT levels [135]. In addition, reduction of oxidative stress by empagliflozin treatment resulted in amelioration of hepatic inflammation [30,107,116] and steatosis, as judged by down-regulation of inflammatory markers [30,107,115]. Of interest, decreased macrophage infiltration expression [107,117,134] as well as elevation of autophagy markers [30,117] have also been linked to the anti-inflammatory actions of SGLT-2-i in the liver. Specifically, oxidative stress was reduced after empagliflozin treatment via up regulation of Nrf2, which led to increased antioxidant enzyme activity (SOD) and reduced hepatic inflammation [115]. Empagliflozin exerted anti-inflammatory actions in diet-induced obese mice and NASH mouse models through, at least in part, suppression of hepatic nuclear factor kappa-light-chain-enhancer of activated B cells (NFκΒ), monocyte chemoattractant protein-1 (MCP-1) and TNF-α expression, as well as inhibition of the IL17/IL23 axis [115,117,136]. Reduced hepatic inflammation has also been reported in mice and rats after treatment with low dose (1 mg/kg/day) dapagliflozin and ipragliflozin. Indeed, ipragliflozin administration for 12 weeks alleviated hepatic fibrosis through reduced expression of pro-inflammatory markers Emr1 and Itgax in non-tumoric lesions [126]. On the contrary, a recent study in mice by Hupa-Breier et al., demonstrated that four weeks treatment of diet-induced NASH with empagliflozin alone at a dose of 10 mg/kg/day did not exert any beneficial effect on NASH, while it significantly increased expression of pro-inflammatory and pro-fibrotic genes. In the same study, the combination of dulaglutide (a GLP-1 agonist) and empagliflozin exhibited a hepato-protective effect on diabetic background mice through modulation of the pro-inflammatory immune response and microbiome dysbiosis [137].

Meng et al., demonstrated that induction of AMPK/mTOR activity is dominant pathway in empagliflozin-induced beneficial effects on liver inflammation and ALT levels [117]. Dapagliflozin treatment also induced AMPK/mTOR pathway activation through phosphorylation of liver kinase B1 (LKB1; a ubiquitously expressed master serine/threonine kinase that activates downstream kinases of the AMPK pathway) [105,138]. Previous studies have demonstrated that AMPK activation is required for SGLT-2i induced autophagy-dependent lipid catabolism [30,117]. Both dapagliflozin and empagliflozin promote hepatic autophagy through increased AMPK phosphorylation and BECLIN gene expression, as well as reduced P62 levels and mTOR levels and activity [30,85].

Apart from impaired autophagy, ER stress is also implicated in the development of steatosis and the progression of NAFLD/NASH [30,107]. Consistent with the regulatory role of ER stress in the autophagy process, Nasiri-Ansari et al., demonstrated that stimulation of autophagy by empagliflozin leads to amelioration of ER stress and reduced hepatic cell apoptosis [30]. In particular, empaglifilozin administration protects against HFD-induced NAFLD through inhibiting all three branches of ER stress, namely inositol-requiring enzyme 1 (IRE1a), X-box binding protein 1 (Xbp1), activating transcription factor 4 (ATF4), C/EBP homologous protein (CHOP) and activating transcription factor 6 (ATF6) [30,107]. A recent study by Chen et al., also showed that down-regulation of the ATF6 signaling diminished ER stress-induced inflammation and apoptosis of hepatic cells [139]. The synergistic ER stress response and autophagy process regulated hepatic cell apoptosis in HFD-fed ApoE^−/−^ mice with NASH, by reducing cleaved caspase 3 levels and elevation of B-cell lymphoma-2 (Bcl-2)/BCL2 Associated X (Bax) ratio [30]. Of note, canagliflozin was previously found to exert an anti-apoptotic effect on HFD-fed mice as revealed by robust Bcl-2 hepatic expression [129].

Chronic liver inflammation leads to transformation of hepatic stellate cells to myofibroblasts, thus contributing to liver fibrosis and progression of NAFLD to NASH [140]. Transforming growth factor beta (TGF-β) is known as the most potent inducer of liver fibrosis. Administration of canagliflozin for 16 and 20 weeks reduced the expression of hepatic TGF-β in both F344 rats and MC4R-KO mice, respectively [101,110]. Interestingly, TGF-β activation leads to increased fibronectin and collagen types I, II, and IV production [141]. Treatment of mice with both empagliflozin and canagliflozin resulted in reduced expression of different types of collagen, implying synergistic action(s) [87,137]. Luseogliflozin also exerts anti-fibrotic effects through reduction of collagen1a1, collagen1a2, TGF-β and smooth muscle actin (SMA) expression [142]. Extensive data exist regarding the crucial role of tissue inhibitors of metalloproteinases (TIMPs) in the progression of hepatic fibrosis. Reduced expression of TIMPs and amelioration of hepatic fibrosis have been observed after luseogliflozin and canagliflozin administration [101,126,142]. On the contrary, empagliflozin treatment did not affect the expression of stellate cell functionality markers, such as galectin-3, a-SMA, and collagen1a1 [99].

NASH has been associated with an increased risk of cirrhosis and HCC development. Hepatic tumorogenesis is the result of DNA instability caused by several factors, such as hepatic lipotoxicity, aberrant metabolism and inflammation [141,143]. As a protective effect of SGLT-2i on NAFLD/NASH progression has been shown in the vast majority of the studies, further research aimed to evaluate the effect of SGLT-2i on development and progression of HCC [87,101,142]. Activation of the hepatic P53/P21 signaling pathway is positively correlated with induced fibrosis and hepatocellular carcinogenesis. A 14-week treatment with tofogliflozin has been shown to effectively reduce the P21 expression, alleviating the progression of NASH through, at least in part, mitigating genes related to the hepatocyte cellular senescence-associated secretory phenotype [89]. Daily administration of canagliflozin prevented the occurrence of HCC in both STAM mice—a NASH model—(11 weeks) and HFD-fed MC4R-KO mice (52 weeks), as indicated by a significantly reduced number of hepatic tumorous lesions [87,101]. Moreover, 66 weeks administration of canagliflozin to CDAA-fed F344 rats exerted an anti-carcinogenic effect, as indicated by a reduced number of cells positively stained for placental glutathione S-transferase (a prominent marker of hepatocarcinogenesis) [110]. Finally, canagliflozin administration reduced the development of hepatic preneoplastic lesions in CDAA-fed rats through suppression of hepatic neovascularization markers, namely cluster of differentiation 31 (CD31) and vascular endothelial growth factor (VEGF) [110]. Overall, the key animal studies regarding the impact of SGLT-2i on NAFLD/NASH/HCC are summarized in Table 1, Table 2, Table 3 and Table 4.

**Table 1 ijms-23-03107-t001:** Key animal studies regarding the impact of empagliflozin on NAFLD/NASH.

Study/Reference	Animal Model, Dose, & Duration	Effect on Body Weight and Liver Weight	Effect on Laboratory Values	Mechanism of Action	Effect on Insulin Sensitivity & Glucose Homeostasis	NAFLD Activity Score (NAS) & Fibrosis/Steatosis
Perakakis, N., et al., 2021 [99]	Male C57BL/6JRj on AMLN diet (HFD, fructose + cholesterol) 10 mg/kg/day 12 (w)	No effect	-	⬇ Hepatic lactosylceramides	⬇ Blood glucose levels No effect on Insulin sensitivity	⬇ Lobular inflammation ⬇ NAS No effect on the hepatic steatosis and fibrosis
Meng, Z., et al., 2021 [117]	Male C57BL/6J on HFD + streptozotocin injection (T2DM with NAFLD) 10 mg/kg/day 8 (w)	⬇ Body weight ⬇ Liver/Bw	⬇ ALT ⬇ TG & TC ⬆ HDL	⬇ Lipogenesis markers and lipid uptake genes (SREBP1, ChREBP, FASN, ACCα, SCDα, CD36) ⬆ Autophagy activation (AMPK/mTOR & BECLIN1, LC3BII) ⬆ IL-17/IL-23 axis inhibition (IL-23p19, IL-23, IL-1β, IL-17A, RORγt, p-STAT3/t-STAT3, IL-17) ⬇ M1 macrophage marker (CD11C, CD86, NOS2) ⬇ Th17-related chemokines and chemokine receptors (CCL20, CCR6, CCR4, CXCL1/CXCL2, CXCL1, CXCR2)	⬇ Blood glucose levels	⬇ Hepatic steatosis ⬇ NAS
Nasiri-Ansari, N., et al., 2021 [30]	Male ApoE knockout mice on HFD 10 mg/kg/day 10 (w)	No effects	⬇ ALT, TG, TC levels ⬇ Serum TG/HDL levels	⬇ Lipogenesis markers (SREBP1, Pck1, FASN) ⬇ Inflammatory markers (MCP-1, F4/80) ⬇ ER stress markers (GRP78, IRE1α, XBP1, ELF2α, ATF4, CHOP, GRP94) Autophagy markers (⬇ mTOR and P62 & ⬆ pAMPK/AMPK, BECLIN) ⬇ Apoptosis markers (Bax/Bcl-2 Ratio, cleaved Caspase-3)	⬇ Blood glucose levels	⬇ Lobular inflammation ⬇ Hepatic steatosis ⬇ NAS No effect on hepatic fibrosis
Petito-da-Silva, et al., 2019 [107]	Male C57Bl/6 mice on HFD 10 mg/kg/day 5 (w)	⬇ BW ⬇ Liver/Bw	No effect on ALT	⬇ Lipogenic genes (Fas, SREBP1c, PPARγ) ⬇ ΕR- stress markers (CHOP, ATF4, GADD45) Fatty acid β-oxidation (⬆ PPAR-α, ⬇ Acox1) ⬇ lipid droplet-associated protein (Fsp27/cidec) ⬇ Inflammatory markers (Nfκb, TNF-α)	Improved Glucose intelorence Improved Insulin sensitivity	⬇ Hepatic TC ⬇ Hepatic steatosis
Jojima, T., et al., 2016 [136]	Male C57BL/6J on HFD + early STZ injection 10 mg/kg/day 3 (w)	⬇ Liver/BW	⬇ GA ⬇ ALT	⬇ Inflammatory markers (IL6, TNF-α, MCP-1, SOCS3) ⬇ Plasma DPP-4 activity (CD26/DPP-4)	⬇ Plasma glucose Levels	⬇ Hepatic TG ⬇ NAS ⬇ Hepatic fibrosis
Hüttl, M., et al., 2021 [115]	HHTg & Wistar rats 10 mg/kg/day 8 (w)	No effect on BW	⬇ TAG No effect on serum ALT	⬇ Lipogenicgenes (Fas, Scd-1, SREBP1c, PPARγ) Improvement of hepatic lipid metabolism (⬆ Nrf, Cyp2e1, ⬇ FGF21, Cyp4a1, Cyp1a1, Cyp2b1) ⬇ Inflammatory markers (MCP-1) ⬆ Oxidative stress markers (⬇ Hepatic GSH/GSSG, SOD) ⬇ Hepatokines (Fetuin-A)	Improved Glucose intolerance Improved Insulin sensitivity	⬇ Hepatic TG ⬇ lipotoxic diacylglycerols ⬇ Fibrosis

**Table 2 ijms-23-03107-t002:** Key animal studies regarding the impact of canagliflozin on NAFLD/NASH/HCC.

Study/Reference	Animal Model Dose & Duration	Effect on Body & Liver Weight	Effect on Laboratory Values	Mechanism of Action	Effect on Insulin Sensitivity & Glucose Homeostasis	NAFLD Activity Score (NAS) & Fibrosis/Steatosis
Yoshino, K., et al., 2021 [111]	obese diabetic KK-Ay mice putative dose of ~17 mg/kg/day 3 (w)	No effect on body weight No effect epididymal fat weigh ⬇ Liver/Bw	⬇ TG No effect on serum ALT	⬆ Prostaglandin E2 (PGE2) and resolvin E3	⬇ Plasma glucose levels	⬇ Hepatic TG
Tanaka K., et al., 2020 [108]	Male C57BL/6J mice on HFD 30 mg/kg/day 4 (w)	⬇ BW	⬇ ALT ⬆ TG, ketone bodies	⬆ lipid-dependent energy expenditure ⬇ Respiratory Qquotients ⬇ Lipogenesis markers (PPAR, FAS, Scd1) ⬆ Fatty acid β-oxidation markers (CPT1a, PGC1a, PGC1b) ⬇ Inflammatory markers (IL-1b)	Improved glucose intolerance Improved insulin sensitivity ⬇ lasma glucose & insulin levels	⬇ Hepatic TG ⬇ NAS
Jojima, T., et al., 2019 [87]	STAM mice 30 mg/kg/day 4 & 11 (w)	⬇ Liver/Bw (11 w)	⬇ TG (11 w) ⬇ ALT (11 w)	⬇ Inflammatory marker and fibrosis marker [SOCS-3, collagen 3 (4 w)] ⬇ Lipogenesis markers [FAS (11 w)] ⬇ Inhibits progression of NASH to Hepatocarcinogenesis GS & AFP	⬇ Plasma glucose levels	⬇ NAS (11 w) ⬇ Hepatic fibrosis (4 w) ⬇ Tumor number (11 w)
Shiba, K., et al., 2018 [101]	MC4R-KO mice on HFD 20–30 mg/kg/day 8, 20 & 52 (w)	⬇ Liver weight (8 w) ⬆ Body weight (8 and 52 w)	⬆ TG (8 and 20 w) ⬇ ALT (8 and 20 w)	⬇ Lipogenic markers genes [Acc1 and Scd1 (8 and 20 w), Fasn (8 w)] ⬇ Gluconeogenic markers (G6pc, Pck1) ⬇ Inflammatory markers [F4/80 gene (20 w), TNFa (20 w), Cd11c (8 and 20 w)] ⬇ Fibrosis markers [Col1a1, TIMP-1 (8 and 20w), Acta2, Tgf1b (20 w)]	Improved insulin sensitivity and hyperglycemia (8 and 20 w)	⬇ Hepatic steatosis (8 w) ⬇ Hepatic fibrosis (20 w) ⬇ NAS (20 w) ⬇ Tumor number (52 w) ⬇ Hepatic TG content (8 and 20 w)
Ozutsumi, T., et al., 2020 [110]	F344 rats on CDAA diet 10 mg/kg/day 16 (w)	No effect on body weight No effect on Liver/BW	⬇ ALT	⬇ Fibrosis markers (αSMA, TGF-β1, α1(I)-procollagen) ⬇ Inflammatory markers (CCL2, TNF-α, IL-6) ⬇ Hepatocarcinogenesis markers (GST-P, VEGF, CD31) ⬇ Oxidative stress markers (MDA, 8-OHdG)	No effect on insulin sensitivity No effect on plasma glucose levels	⬇ Hepatic fibrosis & Steatosis ⬇ Hepatic Cirrhosis ⬇ Hepatic inflammation ⬇ Hepatic ballooning ⬇ NAS
Kabil, Sl, et al., 2018 [129]	Male Wister rats injected with STZ on HFD 10 and 20 mg/kg/day 8 (w)	⬇ Liver weight ⬇ BW (20 mg)	⬇ ALT ⬆ TC, TG & NEFA	⬆ Hepatic lipolytic factor ZAG ⬇ Inflammatory markers (serum TNF-α, hepatic IL-6) Serum apoptotic markers (⬇ Caspase3, ⬆ Bcl-2) Hepatic oxidative stress (⬇ MDA, ⬆ SOD and GPx activity) Serum antioxidant enzyme activity (⬇ TOS and ⬆ TAS)	No effect on fasting insulin levels Fasting blood glucose	⬇ Hepatic inflammation ⬇ Hepatic TC, TG, NEFA ⬇ Hepatic inflammation Hepatic Steatosis ⬇ NAS

**Table 3 ijms-23-03107-t003:** Key animal studies regarding the impact of dapagliflozin on NAFLD/NASH.

Study/Reference	Animal Model Dose & Duration	Effect on Body and Liver Weight	Effect on Laboratory Values	Mechanism of Action	Effect on Insulin Sensitivity & Glucose Homeostasis	NAFLD Activity Score (NAS) & Fibrosis/Steatosis
Han, T., et al., 2021 [105]	Male C57BL/6 J and ob/ob mice on HFD 1 mg/kg/day 4 (w)	No effect on BW	⬇ TC	⬆ β-oxidation (PPAR-α, CPT1, PGC1α) ⬇ Inflammatory markers (MCP1)	⬇ Fasting blood glucose	⬇ Hepatic oxidative stress ⬇ Hepatic lipid accumulation ⬇ Hepatic steatosis
Luo, J., et al., 2021 [84]	Male NIH mice on HFD 25 mg/kg/day 4 (w)	No effect on BW ⬆ Food intake	⬇ ALT	⬇ Lipogenic markers (SREBP1, ACC, FASN) ⬆ β-oxidation markers (PPARα, CPT1a) Regulation of lipid metabolism ⬆ pAMPK and ⬇ pmTOR	-	⬇ Hepatic steatosis ⬇ Hepatic ballooning ⬇ HepaticTC, TG
Tang, L., et al., 2017 [132]	db/db mice 1.0 mg/kg/day via diet gel 4 (w)	No effect on BW	⬇ ALT ⬇ TG	⬇ Inflammatory markers (MPO, F4/80) ⬇ Oxidative stress markers (ROS) ⬇ Fibrosis markers (FN, Col I, Col III, LM)	⬇ Plasma glucose levels	⬇ Hepatic injury ⬇ Hepatic fibrosis ⬇ Hepatic inflammation
Yabiku, K., et al., 2020 [109]	Male C57BL/6J mice on HFD or HFD + MCDD 0.1 or 1.0 mg/kg/day 2 (w)	⬇ BW (0.1 and 1 mg) in both diets	⬇ ALT (0.1 and 1 mg) Mice on HFD	-	Improved glucose tolerance and insulin sensitivity	-
Omori, K., et al., 2019 [125]	db/db mice on ND 1.0 mg/kg/day 8 (w)	No effect on BW ⬇ Liver weight	⬇ TG ⬇ Plasma C-peptide	No significant differences in the expression of fatty acid oxidation markers No significant differences in the expression of inflammatory markers ⬇ Fatty acid uptake and storage markers (PPARγ targeted genes as compared to Gla group)	Improved glucose tolerance	No significant changes in hepatic TG, Palmitate, Oleate, and Stearate content
Li, L., et al., 2021 [85]	ZDF rats 1 mg/kg/day 9 (w)	⬇ BW ⬇ Liver weight ⬇ Liver weight/BW	⬇ TG, TC, LDL, HDL	⬇ Lipogenic markers (SREBP1, ACC1, p-ACC) ⬆ Fatty acid oxidation markers (ACOX1, CPT1, pACOX) Autophagy-related markers (⬆ LC3B, Beclin1, activation of AMPK/mTOR pathway and ⬇ P62)	⬇ Plasma glucose and insulin levels	⬇ Hepatic lipid accumulation ⬇ Hepatic steatosis
ElMahdy, M.K., et al., 2020 [130]	Male Wistar rats on HCHF diet 1 mg/kg/day 5 (w)	No significant effects on liver weight	⬇ ALT, AST ⬇ TC, TG, LDL ⬆ HDL	⬇ Inflammatory markers (TNF-α, IL-1β, IL-18)	-	⬇ Hepatic steatosis

**Table 4 ijms-23-03107-t004:** Key animal studies regarding the impact of ipragliflozin, remogliflozin, tofogliflozin and luseogliflozin on NAFLD/NASH/HCC.

Study/Reference	Animal Model Dose & Duration	Effect on Body Weight &Body Composition	Effect on Laboratory Values	Mechanism of Action	Effect on Insulin Sensitivity & Glucose Homeostasis	NAFLD Activity Score (NAS) & Fibrosis/Steatosis
**Ipragliflozin**						
Tahara, A. & Takasu, T., 2020 [100]	KK-Ay mice on HFD 0.1, 0.3, 1 and 3 mg/kg/day Alone or with Metformin 4 (w)	⬇ BW weight (1 & 3 mg) ⬇ Liver weight (0.3, 1 & 3 mg)	⬇ TG (0.3, 1 & 3 mg) ⬇ TC (1 & 3 mg) ⬇ AST (1 & 3 mg) ⬇ ALT (0.3, 1 & 3 mg)	⬇ Inflammatory Markers [serum TNF-α, IL-6, MCP-1 and CRP (1 and 3mg); Liver TNF-α (3 mg) and IL-6, MCP-1 and CRP (1 & 3 mg)] ⬇ Serum and hepatic oxidative stress markers [TBARS and protein carbonyl (1 & 3 mg)]	Improve glucose intolerance Improved Insulin resistance Improved hyperlipidemia	⬇ Hepatic TG, TC (1 & 3 mg) ⬇ Hepatic Hyperthrophy (1 & 3 mg) ⬇ Hepatic Inflammation (1 & 3 mg) ⬇ Hepatic fibrosis & steatosis (3 mg)
Tahara, A., et al., 2019 [113]	KK-Ay mice on HFD 0.1, 0.3, 1 and 3 mg/kg/day Alone or with Pioglitazone 4 (w)	⬇ BW weight (1 & 3 mg) ⬇ Liver weight (0.3, 1 & 3 mg)	⬇ TC (0.3, 1 & 3 mg) ⬇ TG (1 & 3 mg) ⬇ AST (1 & 3 mg) ⬇ ALT (1 & 3 mg)	⬇ Genes involved in regulation of insulin sensitivity (Plasma adipocytokines, Leptin & FGF-21) ⬇ Inflammatory Markers [serum TNF-α, IL-6, MCP-1 and CRP (1 and 3 mg); Liver TNF-α (3 mg) and IL-6, MCP-1 and CRP (1 and 3 mg)] ⬇ Serum and hepatic oxidative stress markers [TBARS and protein carbonyl (1 & 3 mg)]	⬇ Plasma glucose and insulin levels (0.3, 1 and 3 mg)	⬇ Hepatic TG (0.3, 1 & 3 mg) ⬇ Hepatic TC (1 & 3 mg) ⬇ Hepatic Hyperthrophy (1 & 3 mg) ⬇ Hepatic Inflammation (3 mg) ⬇ Hepatic fibrosis (3 mg)
Komiya, Ch, et al., 2016 [112]	ob/ob and WT mice on HFD 11 mg/kg/day 4 (w)	⬇ Hyperphagia ⬇ BW weight ⬇ Liver weight	⬇ ALT ⬇ TG ⬇ Plasma glucagon	⬇ Lipogenic markers genes (SREBP1, Fasn, Acc1, Scd1) ⬇ Gluconeogenic markers (Pck1) ⬇ Inflammatory markers (F4/80, Cd11c)	Improved Insulin resistance Improved fasting glucose levels	⬇ Hepatic TG ⬇ Hepatic lipid ⬇ Hepatic steatosis
Honda, Y., et al., 2016 [114]	C57BL/6J male mice on AMLN diet 40 mg/kg/day 8 (w)	No effect on BW ⬇ Liver weight	⬇ ALT, AST ⬇ FFA	⬆ β-oxidation (PPAR-α, CPT1, MTTP) ⬇ Hepatocytes apoptosis (TUNEL) ⬇ Lipogenic markers genes (SREBP1 & Acc1)	Improved Insulin resistance	⬇ Hepatic TG & FFA ⬇ Hepatic fibrosis ⬇ Hepatocyte ballooning ⬇ Lobular inflammation ⬇ NAS
Hayashizaki-Someya, Y., et al., 2015 [144]	Male Wistar rats on CDAA diet 0.3 and 3 mg/kg/day 5 (w)	⬇ BW weight (3 mg)	No effect on ALT, AST	-	No effect on fasting blood glucose levels	⬇ Hepatic TG (3 mg) ⬇ Hepatic lipid droplet size ⬇ Hepatic fibrosis (0.3 & 3 mg) ⬇ Hepatic HP
Yoshioka, N., et al., 2021 [126]	Mc4r KO mice on HFD and injected with single dose of diethylnitrosamine 5 mg/kg/day 12 (w)	⬇ BW weight ⬇ Liver weight	⬇ ALT, AST ⬇ LDH	⬇ Lipogenic markers genes (Fasn in non-tumor) ⬇ Fibrosis markers (Emr1, Itgax in non-tumor) ⬇ Cell senescence markers (Cxcl1 in tumor lesion; p21, Cxcl1, MMp12, mmp13 in non-tumor) ⬆ β-oxidation (PPAR-α in tumor lesion; PPAR-α, CPT1, PGC1 in non-tumor) ⬇ Cell apoptosis (Bax and Pcna)	⬇ Plasma glucose & insulin levels	⬇ Hepatic TG ⬇ Lobular inflammation ⬇ Hepatocyte ballooning ⬇ NAS ⬇ Hepatic steatosis & fibrosis ⬇ Hepatic tumor number & size
**(Remogliflozin)**						
Nakano, S., et al., 2015 [134]	C57BL/6J mice on HFD32 13.2 ± 2.2 and 33.9 ± 2.0 mg/kg/day 4 (w)	⬇ Liver weight ⬇ Liver/BW	⬇ ALT & AST	⬇ Inflammatory markers [Hepatic TNF-α (13.2 mg), hepatic MCP-1 (13.2 and 33.9 mg)] ⬇ Oxidative stress (serum and hepatic TBARS)	Improved non fasting glucose levels	⬇ Hepatic TG ⬇ Hepatic fibrosis
**Tofogliflozin**						
Obara, K., et al., 2017 [89]	db/db mice on HFD and injected with single dose of diethylnitrosamine 1 and 10 mg/kg/day 14 (w)	⬇ Liver weight (10 mg)	⬇ ALT (10 mg) ⬇ FFA (1 & 10 mg)	⬇ Inflammatory markers (10 mg) (F4/80)	⬇ Plasma glucose levels Improved insulin insensitivity	⬇ Foci of cellular alteration (10 mg) ⬇ Hepatic pre-neoplastic lesions (10 mg) ⬇ Hepatocyte balooning (10 mg) ⬇ Hepatic steatosis (10 mg) ⬇ NAS (1 & 10 mg)
**Luseogliflozin**						
Qiang, Sh, et al., 2015 [142]	C57BL/6 mice injected with STZ on HFDT Mixing in food at 0.1%. *w*/*w* food 8 (w)	⬇ Liver weight	⬇ ALT ⬆ TG, NEFA	⬇ Hepatic fibrosis markers (collagen1a1, collagen1a2, TGF, SMA, TIMP1) ⬇ Inflammatory markers (MCP-1, IL1, IL-12, IL-6, f4/80)	⬇ Plasma glucose levels	⬇ Hepatic TC, TG &NEFA

Abbreviations: W: week, BW: Body Weight, STZ: streptozotocin, WT: Wild Type, Gla:insulin glargine, ALT: Alanine aminotransferase, AST: Aspartate aminotransferase, TG: Triglycerides, TC: Total cholesterol, FFA: Free fatty acids, GA: glycated albumin, STAT3: Signal Transducer And Activator Of Transcription 3, ROR: RAR Related Orphan Receptor, Bax: BCL2 Associated X, IRE: Inositol-requiring enzyme 1, -Xbp1: X-box binding protein 1, ATF4: Activating transcription factor 4, CHOP: C/EBP homologous protein, ATF-6:Activating transcription factor 6, ChREBP: Carbohydrate response element binding protein, SREBP1: Sterol regulatory element-binding transcription factor 1, Scd1: Stearoyl-CoA desaturase1, ACC1: Acetyl-CoAcarboxylase 1, ACOX: Peroxisomal acyl-CoA oxidase1 Fasn: Fatty acid synthase, IL: Interleukin, TNF-α: Tumor necrosis factor, NFκΒ: Nuclear factor kappa-light-chain-enhancer of activated B cells, MCP-1:Monocyte chemoattractant protein-1, CRP:C-reactive protein, PPAR-α: Peroxisome proliferator activated receptor alpha, CPT1: Carnitine palmitoyltransferase 1, PGC1: PPARγ coactivator 1, TOS: Total Oxidant Status, TAS: Total antioxidant status, p21: cyclin-dependent kinase inhibitor, Cxcl: Chemokine (C-X-C motif) ligand, MMP: Matrix metalloproteinases, TIMP: Tissue inhibitors of matrix metalloproteinases, CYP: Cytochromes P450, MDA: Malondialdehyde.

### 4.2. Human Trials

A number of clinical studies have highlighted the benefit of SGLT-2i in patients with T2DM and NAFLD (Table 5) [145,146,147,148,149,150,151,152,153,154,155,156,157,158,159,160,161,162,163,164,165,166,167,168,169]. In the majority of them, the administration of SGLT-2i has resulted in improvement of serum levels of liver enzymes and hepatic steatosis, as evaluated by magnetic resonance imaging (MRI), ultrasound (U/S), non-invasive biomarkers, such as AST to platelet ratio (APRI) index, NAFLD fibrosis score (NFS) and Fibrosis-4 (FIB-4) score, or even by liver biopsy (LB). In some of these studies, improvement in hepatic fibrosis was found, using transient elastography (TE) or LB, even though this finding was not univocal [146,153,158,159,164,165,168].

Of note, the overall improvement in hepatic steatosis is also found in patients treated with other classes of anti-diabetic drugs, like thiazolidinediones and dipeptidyl peptidase-4 (DPP-4) inhibitors, raising the question of steatosis improvement due to glycemic control [148,160,162,170]. Moreover, the vast majority of the aforementioned studies are limited by the small sample size, heterogenous inclusion criteria, especially regarding the presence of NAFLD, as well as the duration of follow up, thus meta-analyses have been conducted to better assess the true benefit of SGLT-2i in patients with DM and NAFLD [171,172,173]. In the largest one, comprising 9 randomized trials, with 7281 and 4088 patients in the SGLT-2i and control arm, (standard of care (SOC) or placebo), respectively, the use of SGLT-2i resulted in improvement of serum transaminases, body weight, and liver fat as measured by proton density fat fraction. Accordingly, the authors discuss that this improvement derives mainly from the achievement of glycemic control and weight loss. However, in a 2020 study by Kahl et al., including 84 patients with DM and excellent glycemic control, randomly assigned to empagliflozin or placebo, patients on empagliflozin had improved liver fat content, as assessed by MRI [161], strongly suggesting that the good glycemic control and weight loss are not the only mechanisms associated with the beneficial effects of SGLT-2i treatment on hepatic steatosis.

In non-DM patients, only a small single center study exists which studied 12 patients under dapagliflozin and 10 patients under teneligliptin, a DPP4 inhibitor, for a total of 12 weeks, showing that after this intervention period, serum transaminases were decreased in both groups, while in the dapagliflozin group, total body water and body fat decreased, leading to decreased total body weight [174].

Regarding the pathophysiology of NAFLD improvement under SGLT-2i treatment, various mechanisms have been suggested. Treatment with SGLT-2i results in decreases in both glucose and insulin levels (especially in patients with DM), which lead to a large reduction of hepatic de novo lipid synthesis [175]. Moreover, glucagon-secreting alpha pancreatic cells also express SGLT-2, thus the administration of SGLT-2i stimulates glucagon secretion [175,176,177]. In turn, the subsequently elevated plasma glucagon levels stimulate β-oxidation, and this shift from carbohydrate to fatty acid metabolism leads to reduced liver triglyceride content and consequently hepatic steatosis [175,178,179]. Another potential mechanism is mediated by the antioxidant effects of SGLT-2i. Apart from their ability to reduce high glucose-induced oxidative stress, SGLT-2i reduce free radical generation, suppress pro-oxidants, and upregulate antioxidant systems, such as superoxide dismutases (SODs) and glutathione (GSH) peroxidases (Figure 1) [116,180,181,182,183,184].

**Table 5 ijms-23-03107-t005:** Studies on the impact of SGLT-2i on NAFLD in patients with T2DM.

Study	Study Design	No of Pts	SGLT-2i/Drug Used (No of Pts)	Control Group	Treatment Duration (Weeks)	NAFLD Diagnosis **	Key Results
Eriksson, J., et al., 2018 [150]	Randomised, double-blind, prospective	84	Dapagliflozin (42)	OM-3CA or placebo	12	MRI	Reduction of serum transaminases, CK-18, FGF-21 in Dapagliflozin group and liver fat in Dapagliflozin + OM-3CA group
Kahl, S., et al., 2020, [161]	Randomised, double-blind, prospective	84 *	Empagliflozin (42)	Placebo	24	MRI	LFC improvement only in empagliflozin
Chehrehgosha, H., et al., 2021 [165]	Randomised, double-blind, prospective	78	Empagliflozin (21)	Pioglitzone or placebo	24	TE	Better CAP, LS, no difference vs. pioglitzone for serum transaminases or FIB-4
Gaborit, B., et al., 2021 [167]	Randomised, double-blind, prospective	34	Empagliflozin (18)	Placebo	12	MRI	Reduction in liver fat vs. placebo
Bando, Y., et al., 2017 [145]	Randomised, open label, prospective	62	Ipragliflozin (40)	SOC	12	C/T	Improvement in serum transaminases. VFA, L/S ratio compared to SOC
Ito, D., et al., 2017 [147]	Randomized, open label, prospective	66	Ipragliflozin (32)	Pioglitazone	24	C/T or U/S	Improvement of L/S ratio, ALT, ferritin not statistically significant between 2 groups; ipragliflozin more weight and VFA reduction
Kuchay, M.S., et al., 2018 [152]	Randomized, open label, prospective	42	Empagliflozin (22)	SOC	20	MRI	Reduction of liver fat and ALT
Shibuya, T., et al., 2018 [154]	Randomized, open label, prospective	32	Luseogliflozin (16)	Metformin	26 (6 months)	C/T or U/S	Improvement in L/S ratio compared to baseline
Shimizu, M., et al., 2019 [155]	Randomized, open label, prospective	57	Dapagliflozin (33)	SOC	24	U/S	Improvement of CAP and LS, especially for high LS at the trial beginning
Han, E., et al., 2020 [160]	Randomized, open label, prospective	44	Ipragliflozin (+metformin +pioglitazone) (29)	Metformin + pioglitazone	24	U/S	Better FLI, CAP, NAFLD liver fat score
Kinoshita, T., et al., 2020 [162]	Randomized, open label, prospective	98	Dapagliflozin (32)	Pioglitazone (33) Glimepiride (33)	28	C/T	Improvement of L/S ratio and ALT with pioglitazone and dapagliflozin
Takahashi, H., et al., 2021 [168]	Randomized, open label, prospective	55	Ipragliflozin (27)	SOC, except pioglitazone, GLP1	72	LB	Statistically significant improvement in NASH resolution and fibrosis improvement in SGLT-2i vs. SOC
Yoneda, M., et al., 2021 [169]	Randomized, open label, prospective	40	Topogliflozin (21)	Pioglitzone	24	MRI	Decrease of liver steatosis in both groups, body weight decrease in topogliflozin
Arai, T., et al., 2021 [164]	Open label, Prospective	100	Canagliflozin (29) Ipragliflozin (12) Tofogliflozin (6) Dapagliflozin (4) Luseogliflozin (4) Empagliflozin (1)	SOC	48	U/S	Decrease in LS and CAP in SGLT-2i during treatment, statistically significant decrease in SGLT-2i vs SOC in ALT, FIB-4
Akuta, N., et al., 2017 [146]	Single-arm, Prospective	5	Canagliflozin (5)	N/A	24	LB	Improvement of NAS score, liver steatosis; fibrosis improvement in 2 pts
Itani, T., et al., 2018 [151]	Single arm, Prospective	35	Canagliflozin (35)	N/A	26 (6 months)	U/S	Improvement in ALT, ferritin, FIB-4 at 3 and 6 months
Miyake, T., et al., 2018 [153]	Single arm, Prospective	43	Ipragliflozin (43)	N/A	24	12 LB, 41 U/S	Reduction in serum transaminases, CAP, not statistically significant reduction in fibrosis
Sumida, Y., et al., 2019 [156]	Single-arm, Prospective	40	Luseogliflozin (40)	N/A	24	U/S	Reduction in transaminases, serum ferritin and liver fat in MRI
Akuta, N., et al., 2019 [157]	Single arm, Prospective	9	Canagliflozin (9)	N/A	24	LB	Histological improvement in all patients
Akuta, N., et al., 2020 [159]	Single arm, Prospective	7	Canagliflozin (7)	N/A	24	LB	Histopathological improvement at 24 weeks sustained to >1 year, transaminases and ferritin better at 24 weeks
Seko, Y., et al., 2017 [148]	Retrospective	45	Canagliflozin (18) Ipragliflozin (6)	Sitagliptin	24	LB	Significant decrease in serum transaminases with both drugs, not statistically significant between SGLT-2i and sitagliptin
Choi, D.H., et al., 2018 [149]	Retrospective	102 (all abnormal ALT)	Dapagliflozin + Metformin (50)	DPP4 + Metformin	44.4 ± 18.4 for dapagliflozin and 50.4 ± 21.6 for DPP4	U/S	Statistically significant decrease in dapagliflozin vs. DPP4
Yamashima, M., et al., 2019 [158]	Retrospective	22	Ipragliflozin (18) Dapagliflozin (2) Tofogliflozin (1) Empagliflozin (1)	N/A	52 (22 pts) and 104 (15 pts)	12 LB, 10 U/S	Lower serum transaminases levels at 12 and 24 months, better CAR and shear wave velocity at 12 months
Yano, K., et al., 2020 [163]	Retrospective	69	Dapagliflozin (10) Canagliflozin (7) Ipragliflozin (3) Empagliflozin (2)	SOC	162	LB	Improvement of serum transaminases in both groups (No head to head comparison)
Euh, W., et al., 2021 [166]	Retrospective	283	Dapagliflozin (58) Empagliflozin (34) Ipragliflozin (3)	SOC, except GLP-1 and Insulin	39	U/S	Statistically significant reduction in ALT and body weight in SLT2i vs. SOC

* All patients with excellent glycemic control. ** Test used to diagnose/assess NAFLD.Abbreviations: RCT: Randomised controlled trial, L/S ratio: Liver to spleen ratio, VFA: Visceral fat area, C/T: Computed tomography, MRI: Magnetic Resonance Imaging, OM-3CA: omega-3 carboxylic acids, LFC: Liver fat content, FIB-4: Fibrosis-4 index, ALT: Alanine aminotransferase, FLI: Fatty liver index, CAP: Controlled attenuation parameter, SGLT-2i: Sodium-glucose co-transporter type-2 inhibitors, NAS score: NAFLD Activity Score, LB: Liver biopsy, GLP-1: Glucagon-like peptide-1, pts: patients.

## 5. Conclusions

There is increasing interest regarding the promising effect(s) of SGLT-2i for the treatment of NAFLD, regardless of the co-existence of T2DM. In addition to weight loss, the beneficial effect(s) of SGLT-2i on NAFLD development and progression appear to be mediated directly through regulation of multiple processes, including ER stress, oxidative stress, low-grade inflammation, autophagy and apoptosis, as revealed by in vitro, animal and clinical studies. Moreover, the observed different effects between members of the SGLT-2i class suggest that there are features specific to individual drugs of this class regarding the underlying mechanism(s) of action and their corresponding effects on NAFLD.

## Figures and Tables

**Figure 1 ijms-23-03107-f001:**
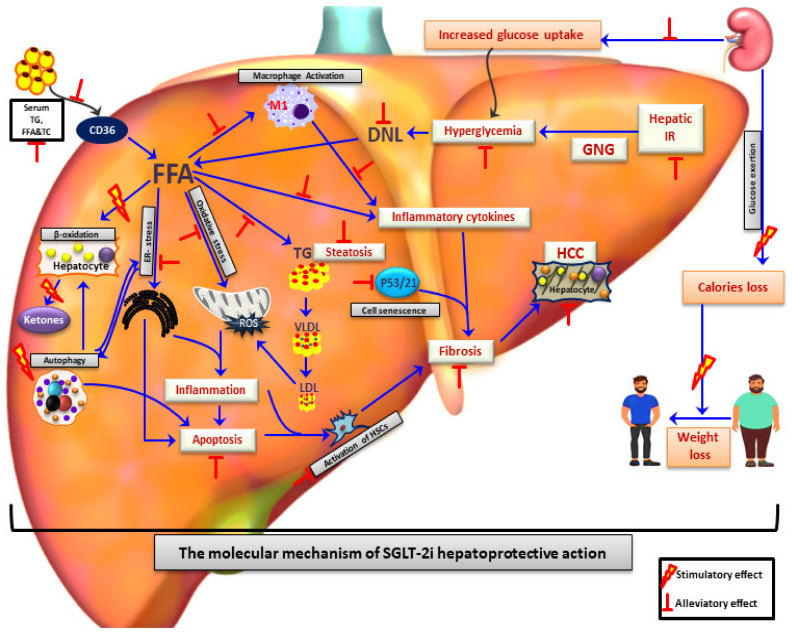
SGLT-2 inhibitors as a promising therapeutic agents for treatment of NAFLD/NASH patients. SGLT-2i treatment contributes to alleviation of NAFLD by reduction of hyperglycaemia, improvement of systematic insulin resistance, elevation of caloric loss and reduction of body weight mostly due to glycosuria. Apart from that, SGLT-2i play a hepatoprotective effect through reduction of hepatic de novo lipogenesis, hepatic inflammation, apoptosis, ER-stress, oxidative stress, and increase of hepatic beta-oxidation. Reduced activation of hepatic satellite cells and p53/p21 pathways by SGLT-2i leads to amelioration of hepatic fibrosis and HCC development. FFA: Free fatty acids; DNL: De novo lipogenesis; HCC: Hepatocellular carcinoma; TC: Total cholesterol; TG: Triglycerides; LDL: Low density lipoprotein; VLDL: Very low density lipoprotein; GNG: Gluconeogenesis; HSC: Hepatic stellate cells; IR: Insulin resistance; ROS: Reactive oxygen species; ER-stress: Endoplasmic reticulum stress.

**Figure 2 ijms-23-03107-f002:**
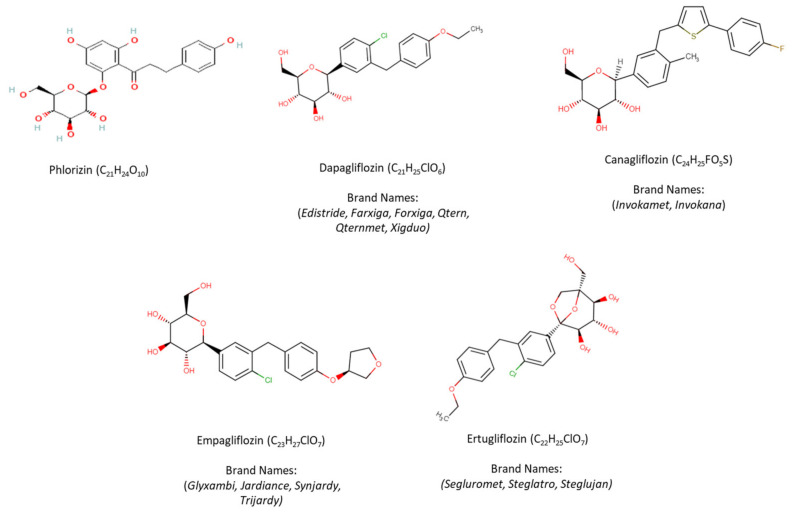
Structure of phlorizin and FDA-approved SGLT-2 inhibitors.

## Data Availability

Not applicable.

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
