# Peer review of "SGLT-2 Inhibitors in NAFLD: Expanding Their Role beyond Diabetes and Cardioprotection"

_ijms, 2022, doi:10.3390/ijms23063107_

Round 1

Reviewer 1 Report

The review deals with a topic of great interest, is well written, very detailed and with a complete and appropriate bibliography.
It would be even more attractive if it were accompanied by a figure representing the pathogenesis of NAFLD including the sites of action of the SGLT-2 inhibitor.

Author Response

Reviewer 1.

The review deals with a topic of great interest, is well written, very detailed and with a complete and appropriate bibliography.
It would be even more attractive if it were accompanied by a figure representing the pathogenesis of NAFLD including the sites of action of the SGLT-2 inhibitor.

Response: We would like to thank the reviewer #1 for his/her valuable suggestion. We have now added Figure 1 to the main body of text accordingly. Cited in page 6, Line 202.

Figure 1. SGLT2 inhibitors as a promising therapeutic agents for treatment of NAFLD/NASH patients. SGLT-2i treatment contributes to alleviation of NAFLD by reduction of hyperglycaemia, improvement of systematic insulin resistance, elevation of caloric loss and reduction of body’s weight mostly due to glycosuria. Apart from that, SGLT-2i play a hepatoprotective effect through reduction of hepatic de novo lipogenesis, hepatic inflammation, apoptosis, ER-stress, oxidative stress, and increase of hepatic beta-oxidation. Reduced activation of hepatic stellate cells and p53/p21 pathways by SGLT2i leads to consequence amelioration of hepatic fibrosis and HCC development.

FFA: Free fatty acids; DNL: De novo lipogenesis; HCC: Hepatocellular carcinoma; TC: Total cholesterol; TG: Triglycerids; LDL: Low density lipoprotein; VLDL: Very low density lipoprotein; GNG: Gluconeogenesis; HSC: Hepatic stellate cells; IR: Insulin resistance; ROS: Reactive oxygen species; ER-stress: Endoplasmic reticulum stress.

Reviewer 2 Report

The author has collected and sorted out many references, but there are problems in the discussion. Firstly, it is not recommended to discuss it according to human test, in vitro test and animal test. It should be discussed according to the specific function and mechanism of sglt-2i..

  1. The chemical formula and structure of sglt-2i shall be given and introduced in detail
  2. Since it is aimed at NAFLD, then SGLT-2i's introduction to cardiovascular disease, diabetes and kidney disease can be deleted.
  3. There is a strong weight loss effect in animal experiments, although some studies have not been reported, which can not be concluded 371-372 “SGLT-2i administration improves biomarkers and/or indicators of liver function and NAFL/NASH injury independently of weight loss”.
  4. The form is too complex and takes up too much space, which is not conducive to reading and refining.

Author Response

Reviewer 2.

The author has collected and sorted out many references, but there are problems in the discussion. Firstly, it is not recommended to discuss it according to human test, in vitro test and animal test. It should be discussed according to the specific function and mechanism of sglt-2i.

Response: We would like to thank the reviewer #2 for this general comment and for his/her time for consideration of our manuscript. His/her thoughtful comments helped us refine the manuscript and improve its’ structure and presented data.

Considering the organization of presented data, we would like to highlight that NAFLD is a complex disease implicated various mechanisms and molecular pathways which closely interact/interplay with each other. Due to this overlap in the molecular mechanisms, the repetition of information would be inevitable in case we organize our manuscript according to the specific function and mechanism of SGLT-2i. Additionally, the exact SGLT-2i mechanism of action in NAFLD/NASH is not yet fully explored, and data regarding its mechanism of action comes mostly from the in vitro and animal studies rather than human trials. Moreover, with this review article we aimed to satisfy the interest of the vast majority of our readers, both clinician and basic research scientists, and provide them with the latest updates in their field of interest at the shortest possible time. Therefore, we thought to organize the section as it is presented in the text in order to have an easy to access without getting lost into information which is not of their individuals’ interest.

Given that, we believe for our readers it would be easier to follow the text if presented according to in vitro, animal and human data. 

1. The chemical formula and structure of sglt-2i shall be given and introduced in detail

Response: We would like to thank the reviewer #2 for this delightful comment.  We have now added the following part to the text’s body. We also added the chemical formula and structures of Phlorizin and currently FDA-approved SGLTSGLT-2i (Figure 2) in the text as follows:

‘’ 3. SGLT-2i overview     

Phlorizin, a b-Dglucoside, was the first natural non-selective SGLT-2i isolated from the bark of the apple tree in 1835 and was initially used for fever relief and treatment of infectious diseases [62,63]. Phlorizin contains a glucose moiety and an aglycone in which two aromatic carbocycles are joined by an alkyl spacer. Approximately fifty years later, scientists observed that phlorizin in high dose lowered plasma glucose levels through glucosuria independently of insulin secretion [62-64]. However, the mechanism of phlorizin action through the active glucose transport system of the proximal tubule was revealed in the early 1970s [62,65].

Thereafter, researchers began to discover the exact mechanism of phlorizin action and its potential application in the management of hyperglycaemia. Phlorizin when delivered orally undergo hydrolysis of the O-glycosidic bond by intestinal glucosidases, leading to formation of phloretin which is capable of GLUT transporters inhibition and uncoupling of oxidative phosphorylation. Due to Phlorizin low SGLT-2 selectivity,  rapid degradation,  inhibition of the ubiquitous glucose transporter 1 (GLUT1), poor intestinal absorption and its gastrointestinal side effects (e.g. diarrhea and dehydration) possibly due to its higher potency for SGLT-1, phlorizin failed to be developed as an antidiabetic drug [62,66-68].

Nevertheless, in order to overcome the aforementioned side effects of phlorizin, researcher developed a devphlorizin-based analogs with improved bioavailability/stability and more selectivity towards SGLT-2 than SGLT-1. Initially, they focused on the development of O-glucoside analogs such as T-1095, sergliflozin and remogliflozin [69,70]. However, due to their poor stability and incomplete SGLT-2 selectivity, pharmaceutical industries turned into development of another derivative of phlorizin known as C-glucosides [62].

Since then, there have been numerous attempts for synthesis of phlorizin C-glucoside analogs with sufficient potency and selectivity for inhibition of SGLT-2. These attempts led to development of dapagliflozin by Meng et al in 2008 [71] . Dapagliflozin demonstrated more than 1200-fold higher potency for human SGLT-2 versus SGLT-1. Apart from dapagliflozin, during the next years several other C-glucoside inhibitors have been developed and approved by United States Food and Drug Administration (FDA). Canagliflozin, characterized by a thiophene derivative of C-glucoside, showed over 400-fold difference in inhibitory activities between human SGLT-2 and SGLT-1 [72]. Thereafter, empagliflozin which has the highest selectivity for SGLT-2 over SGLT-1 (approximately 2700-fold) was the third agent being approved by both the European Medicines Agency (EMA) and FDA while other SGLT-2i namely luseogliflozin and topogliflozin are approved so far for use only in Japan andipragliflozin in Japan and Russia [73-77].

The structure of phlorizin and currently FDA approved  SGLT-2 inhibitors in the United States include dapagliflozin, canagliflozin, empagliflozin, and  ertugliflozin along with their chemical formoula and brand’s name are presented in Figure . Diverse structures of other SGLT-2 inhibitors have been previously disclosed in various articles and in a number of patents [78,79]. Pages 7-9, Lines 214-250.

Figure 2. Structure of phlorizin and FDA-approved SGLT-2 inhibitors.

2. Since it is aimed at NAFLD, then SGLT-2i's introduction to cardiovascular disease, diabetes and kidney disease can be deleted.

Response: We would like to thank the reviewer #2 for this thoughtful comment. We have now omitted these sections.

3.There is a strong weight loss effect in animal experiments, although some studies have not been reported, which cannot be concluded 371-372 “SGLT-2i administration improves biomarkers and/or indicators of liver function and NAFL/NASH injury independently of weight loss”.

Response: We would like to thank Reviewer #2 for his/her thoughtful and detailed comments on our manuscript. Now we corrected the aforementioned text and change it to:

‘’ Although there are studies showing that administration of SGLT-2i does not exert any significant effect on body weight gain [30,99,104,110-114], the vast majority of the literature points towards a beneficial effect on weight loss.

However, the protective role of SGLT-2-i against NAFLD progression seems to also be mediated by effects beyond those on body weight.’’ Page 13, Lines 343-347.

4.The form is too complex and takes up too much space, which is not conducive to reading and refining

Response: We would like to thank the reviewer #2 for this delightful comment. Now, we added a new figure in the text showing the underlying mechanisms of NAFLD pathogenesis as well as the SGLT-2 inhibitor site of action. We believe that our readers will have now a conductive reading, accompanying the text data with the visualized picture. Cited in page 6, Line 202.

We have also shortened our manuscript by omitting three sections (SGLT-2i in cardiovascular disease, diabetes and kidney disease).

Moreover, we split the Table 2 into 4 tables according the specific SGLT-2i drug tested in each animal study. Cited in page 12, Lines 329, 331, 333 and 338.

Figure 1. SGLT2 inhibitors as a promising therapeutic agents for treatment of NAFLD/NASH patients. SGLT-2i treatment contributes to alleviation of NAFLD by reduction of hyperglycaemia, improvement of systematic insulin resistance, elevation of caloric loss and reduction of body’s weight mostly due to glycosuria. Apart from that, SGLT-2i play a hepatoprotective effect through reduction of hepatic de novo lipogenesis, hepatic inflammation, apoptosis, ER-stress, oxidative stress, and increase of hepatic beta-oxidation. Reduced activation of hepatic stellate cells and p53/p21 pathways by SGLT2i leads to consequence amelioration of hepatic fibrosis and HCC development.

FFA: Free fatty acids; DNL: De novo lipogenesis; HCC: Hepatocellular carcinoma; TC: Total cholesterol; TG: Triglycerids; LDL: Low density lipoprotein; VLDL: Very low density lipoprotein; GNG: Gluconeogenesis; HSC: Hepatic stellate cells; IR: Insulin resistance; ROS: Reactive oxygen species; ER-stress: Endoplasmic reticulum stress.

Reviewer 3 Report

I enjoyed reviewing this manuscript. It is interesting and quite well written. This reviewer raises only a few issues that need to be addressed.

1- Introduction. The authors point out that the term NALFD encompasses a wide range of liver diseases ranging from simple steatosis to cirrhosis and HCC. Furthermore, to underline the close pathophysiological relationship of NAFLD with metabolic syndrome / type 2 diabetes, they report the recent use of the term MAFLD. Indeed, it was recently observed by liver biopsy that steatohepatitis represents the sole feature of liver damage in type 2 diabetes. In other words, NAFLD generally presents as NASH in type 2 diabetic patients (PLoS One. 2017 Jun 1;12(6):e0178473. doi: 10.1371/journal.pone.0178473.). This intriguing issue and above reference should be commented.

2- Chapter 2. Overview of NAFLD pathogenesis. Pathophysiological mechanisms linking NAFLD to metabolic syndrome and increased CV risk have recently been presented in some updated reviews (Antioxidants (Basel). 2021 Feb 10;10(2):270. doi: 10.3390/antiox10020270.- Rev Cardiovasc Med. 2021 Sep 24;22(3):755-768. doi: 10.31083/j.rcm2203082.). Above references should be added in the text.

3- Chapter 4. SGLT2 inhibitors in NAFLD. In this chapter the paragraphs on laboratory experiments (in vitro and in vivo) should precede the paragraph on human experiments.

4- Table 2 is too long (7 pages...!). It should be formatted in a more compact format or split into multiple tables.

Author Response

Reviewer 3.

I enjoyed reviewing this manuscript. It is interesting and quite well written. This reviewer raises only a few issues that need to be addressed.

Response: We would like to thank the reviewer #3 for his/her time for consideration of our manuscript. His/her thoughtful comments helped us to improve our article.

1- Introduction. The authors point out that the term NALFD encompasses a wide range of liver diseases ranging from simple steatosis to cirrhosis and HCC. Furthermore, to underline the close pathophysiological relationship of NAFLD with metabolic syndrome / type 2 diabetes, they report the recent use of the term MAFLD. Indeed, it was recently observed by liver biopsy that steatohepatitis represents the sole feature of liver damage in type 2 diabetes. In other words, NAFLD generally presents as NASH in type 2 diabetic patients (PLoS One. 2017 Jun 1;12(6):e0178473. doi: 10.1371/journal.pone.0178473.). This intriguing issue and above reference should be commented.

Response: We would like to thank Reviewer #3 for his/her pointful suggestions. We now added following part into the introduction section along with corresponding reference:

‘’ Masarone et al. revealed that the prevalence of NAFLD was 94.82% among patients with metabolic syndromes (MS) and was presented in all T2DM patients with elevated transaminases when they performed biopsy [12].Surprisingly, 58.52% of MS and 96.82% of T2DM patients in this study were diagnosed with NASH.  As Insulin resistance (IR) is crucially linked with both T2DM and NASH pathophysiology [13], they concluded that NASH may be one of the early clinical manifestations of T2DM [12].’’. Page 4, Lines 126-131.

2-  Chapter 2. Overview of NAFLD pathogenesis. Pathophysiological mechanisms linking NAFLD to metabolic syndrome and increased CV risk have recently been presented in some updated reviews (Antioxidants (Basel). 2021 Feb 10;10(2):270. doi: 10.3390/antiox10020270.- Rev Cardiovasc Med. 2021 Sep 24;22(3):755-768. doi: 10.31083/j.rcm2203082.). Above references should be added in the text.

Response: We would like to thank Reviewer #3 for his/her thoughtful suggestions. Indeed, this part was missing from our manuscript. We now added the following text and the corresponding references to the manuscript:

‘’Of note, NAFLD, particularly steatohepatitis, it has also been associated with an increased risk of cardiovascular-related mortality regardless of age, sex, smoking habit, cholesterolemia and the remaining elements of MS [11]. Several prospective, meta-analyses, observational and cross-sectional studies demonstrated that NAFLD is associated with enhanced preclinical atherosclerotic damage as well as coronary, cerebral and peripheral vascular events with a negative impact on patients’ outcome [14]. Furthermore, the severity of liver biopsy-based fibrosis has been independently associated with the worsening of both systolic and diastolic cardiac dysfunction [15,16].’’  Page 4, Lines 131-137.

3- Chapter 4. SGLT2 inhibitors in NAFLD. In this chapter the paragraphs on laboratory experiments (in vitro and in vivo) should precede the paragraph on human experiments.

Response: We would like to thank the reviewer #3 for this absolutely correct and relevant observation/suggestion. We have now changed the order of presenting data and they are arranged in the text as:

  1. SGLT-2 inhibitors in NAFLD (Page 9, Line 252),

4.1. Laboratory experiments (Page 9, Line 253)

4.1.1. In vitro data (Pages 9-11, Lines 254-306)

4.1.2. Animal studies (Pages 11-27, Lines 307-542)

4.2. Human trials (Pages 27-31, Lines 544-596)

4- Table 2 is too long (7 pages...!). It should be formatted in a more compact format or split into multiple tables.

Response: We would like to thank Reviewer #3 for his/her thoughtful suggestions. Indeed, the table was too long and as you suggested we now split it into 4 different tables according to SGLT-2i used in each group of studies. Cited in page 12, Lines 329, 331, 333 and 338.

Table 1. Key animal studies regarding the impact of empagliflozin on NAFLD/NASH. (Page   19)

Table 2. Key animal studies regarding the impact of canagliflozin on NAFLD/NASH/HCC. (Page 21)

Table 3. Key animal studies regarding the impact of dapagliflozin on NAFLD/NASH. (Page 22)

Table 4. Key animal studies regarding the impact of ipragliflozin, remogliflozin, tofogliflozin and luseogliflozin on NAFLD/NASH/HCC. (Page 24)

Round 2

Reviewer 2 Report

After the major revision the article could be accepted for publishing in IJMS